# HLA-B*27 Heavy Chain Homo-Oligomers Promote the Cytotoxicity of NK Cells via Activation of PI3K/AKT Signaling

**DOI:** 10.3390/medicina58101411

**Published:** 2022-10-07

**Authors:** Hui-Chun Yu, Kuang-Yung Huang, Ming-Chi Lu, Hsien-Yu Huang Tseng, Su-Qin Liu, Ning-Sheng Lai, Hsien-Bin Huang

**Affiliations:** 1Department of Medical Research, Dalin Tzu Chi Hospital, Buddhist Tzu Chi Medical Foundation, Chiayi 62247, Taiwan; 2Division of Allergy, Immunology and Rheumatology, Department of Medicine, Dalin Tzu Chi Hospital, Buddhist Tzu Chi Medical Foundation, Chiayi 62247, Taiwan; 3School of Medicine, Tzu Chi University, Hualien 970, Taiwan; 4Department of Biomedical Sciences, National Chung Cheng University, Chiayi 621, Taiwan

**Keywords:** ankylosing spondylitis, human leukocytic antigen-B*27, NK cells, KIR3DL2, PI3K/AKT signaling

## Abstract

*Background* and *Objectives:* Ankylosing spondylitis (AS) is a chronic inflammatory disease and is highly linked with the expression of the human leukocytic antigen-B*27 (HLA-B*27) genotype. HLA-B*27 heavy chain (B*27-HC) has an innate characteristic to slowly fold, resulting in the accumulation of the misfolded B*27-HC and the formation of homo-oligomeric B*27-HC molecules. The homo-oligomeric B*27-HC can act as a ligand of KIR3DL2. Interaction of the homo-oligomeric B*27-HC molecules with KIR3DL2 will trigger the survival and activation of KIR3DL2-positive NK cells. However, the effects of homo-oligomeric B*27-HC molecules associated with KIR3DL2 on the cytotoxic activity of NK cells and their cytokine expressions remain unknown. *Materials and Methods:* HLA-B*-2704-HC was overexpressed in the HMy2.C1R (C1R) cell line. Western blotting and quantitative RT-PCR were used to analyze the protein expression and cytokine expression, respectively, when C1R-B*-2704 cells that overexpress B*2704-HC were co-cultured with NK-92MI cells. Flow cytometry was used to analyze the cytotoxicity mediated by NK-92MI cells. *Results:* Our results revealed that NK-92MI cells up-regulated the expression of perforin and enhanced the cytotoxic activity via augmentation of PI3K/AKT signaling after co-culturing with C1R-B*2704 cells. Suppression of the dimerized B*27-HC formation or treatment with an inhibitor of PI3K, LY294002, or with an anti-B*27-HC monoclonal antibody can reduce the perforin expression of NK-92MI after co-culturing with C1R-B*-2704. Co-culturing with C1R-B*-2704 cells suppressed the TNF-α and IL6 expressions of NK-92MI cells. *Conclusion:* Stimulation of NK cell-mediated cytotoxicity by homo-oligomeric B*27-HC molecules may contribute to the pathogenesis of AS.

## 1. Introduction

Ankylosing spondylitis (AS) is an autoimmune disease. The characteristics of AS are inflammatory back pain and asymmetric peripheral oligoarthritis [1,2,3,4]. The development of AS is highly associated with the expression of human leukocytic antigen-B*27 (HLA-B*27) [5,6]. More than 90% of AS patients express HLA-B*27. HLA-B*27 is a member of the major histocompatibility complex (MHC) class I molecules that consist of a heavy chain (α chain) and β_2_-microglobulin (β_2_m). HLA-B*27 is assembled with an antigenic peptide in the endoplasmic reticulum (ER). This heterotrimeric complex (HLA-B*27 heavy chain/β_2_m/peptide) can leave the ER and be translocated to the cell membrane for antigen presentation to CD8^+^ T cells [7].

The HLA-B*27 heavy chain (B*27-HC) has a propensity to fold slowly in the ER before it is assembled with β_2_m and an antigenic peptide, leading to a build-up of the misfolded B*27-HC and forming the disulfide-linked heavy chain homo-oligomeric B*27-HC molecules, which can be displayed on the cell surface [8,9,10,11,12,13,14]. It has been known that the homo-oligomeric B*27-HC molecules can act as one of the killer cell Ig-like receptor (KIR3DL2) ligands [15,16,17,18]. In addition, KIR3DL2 is present on the membranes of natural killer (NK) cells and T helper 17 cells (Th17). Recent studies have demonstrated that homo-oligomeric B*27-HC molecules displayed on the membrane of antigen-presenting cells can promote the survival, growth, and IL-17 expression of KIR3DL2-positive CD4 T cells. Although KIR3DL2-positive cells consist of 15% of CD4 T cells in PBMCs, this subgroup accounts for the major numbers of Th17 in AS patients [17,18]. Therefore, the homo-oligomeric B*27-HC molecules are capable of stimulating IL-17 production by Th17 cells and promoting a pro-inflammatory reaction. A monoclonal antibody, HD6, specifically recognizes the B27*-HC homodimer, (B*27-HC)_2,_ but it fails to have affinities with the classic HLA-B*27 heterotrimer (B*27-HC/β_2_m/peptide) [19]. HD6 can disrupt the binding of (B*27-HC)_2_ to KIR3DL2 and inhibit the survival and growth of KIR3DL2-positive NK cells [19]. HD6 suppresses the IL-17 production of PBMCs isolated from AS patients. Taken together, these events possibly provide a strong linkage between the presence of B*27-HC homo-oligomers and the pathogenesis of AS [19].

KIR3DL2 is a homodimeric membrane protein. Each monomer contains three extracellular immunoglobulin-like domains and the long length of the cytoplasmic domain (L representing the long tail). The cytoplasmic domain of each KIR3DL2 monomer contains two immunoreceptor tyrosine-based inhibitory motifs (ITIM) [20]. It has been suggested that the ligand binding to the extracellular domain of KIR3DL2 induces phosphorylation on the tyrosine residues in the ITIM of the cytoplasmic domain, in turn resulting in recruitment and activation of SH2 domain-containing protein tyrosine phosphatase 1 (SHP1) to catalyze the dephosphorylation of phospho-tyrosine and block the signaling induced by protein tyrosine kinase, inhibiting the NK cell-mediated target cell lysis [20]. In addition, it has been reported that phosphorylation of ITIM can also recruit and activate the PI3K/AKT pathway to counter apoptosis, contributing to the signals for NK and T cell growth and survival [21].

KIR3DL2 belongs to a member of the MHC class I-specific inhibitory receptors on NK cells. The major function of KIR3DL2 is to recognize its ligands, the MHC class I molecules, on target cells and block NK cell-mediated target cytolysis [21,22]. However, the effect of homo-oligomeric B*27-HC molecules interacting with KIR3DL2 on the cytotoxic activity of NK cells needs to be characterized. HMy2.C1R (C1R) cells belong to the B-lymphoblast cells that deficiently express the HLA-A* and -B* genes and are sensitive to NK-mediated cell lysis [23]. Transfection of HLA-A*3, -B*7, or -B*58 into C1R can reduce the cytotoxicity mediated by NK cells [23]. In this study, we examined the effect of C1R cells transfected with B*2704-HC or B*27-HC C67S on the NK-mediated cytotoxicity and cytokine expressions of NK cells.

## 2. Materials and Methods

### 2.1. Stable Over-Expression of HLA-B*2704 C67S in HMy2.C1R Cells

Site-directed mutagenesis used to replace Cys-67 of HLA-B*2704 by Ser was carried out by using the QuikChange Site-directed Mutagenesis Kit (Stratagene, La Jolla, CA, USA) with the primer (5′-CGG GAG ACA CAG ATC TCA AAG GCC AAG GCA CGA-3′) and with the template, the pTRE2hyg-HLA-B*2704 vector [24], following the methods described by the manufacturer. The resulting vector was linearized by digest with BamHI and transfected into HMy2.C1R cells via electroporation (Gene Pulser Xcell, BioRad Laboratories, Richmond, CA, USA). Transfected cells were cultured in IMDM with 5% CO_2_, 10% FBS, and hygromycin B (200 μg/ml) (Invitrogen) at 37 ℃. After drug selection, the surviving cells expressing HLA-B*2704 C67S were isolated and analyzed through flow cytometry, following the methods as described [24,25]. 

### 2.2. Biotinylation of Cell Surface Proteins and Pulldown by Avidin Beads

Proteins on the cell surface were biotinylated, following the method as described [26]. Briefly, C1R-B*2704 or C1R-B*2704 C67S cells (3 × 10^6^ cells) resuspended in PBS (1 ml) were reacted with Sulfo-NHS-Biotin (110 mg) (Thermo Fisher Scientific, Waltham, MA, USA) for 30 min with a gentle rotation at 4 °C. The reaction was quenched through the addition of 1 ml of 1 M Tris-HCl buffer, pH 7.4. After reacting for 30 min, the modified cells were isolated through centrifugation (1200 rpm) for 3 min. The pelleted cells were ruptured by the cold RIPA buffer (1 ml). The extracted lysate was incubated with the avidin–agarose beads (60 μl) that were pre-washed with cold PBS. All components were incubated for 2 h with a gentle rotation at 4 °C. After centrifugation at 3000 *g* for 3 min at 4 °C, the precipitated proteins were separated via SDS-PAGE (12.5% acrylamide) and electro-transferred to the PVDF membrane. The bound proteins on the PVDF membrane were detected via the anti-HLA-B*27 heavy chain monoclonal antibody, BH2 [24,25].

### 2.3. Analysis of the Perforin Production of NK-92MI Co-Cultured with HMy2.C1R or C1R-B*2704 or C1R-B*2704 C67S Cells

NK-92MI cells (1 × 10^6^ cells) were co-cultured with C1R-B*2704 cells (5 × 10^5^ cells), C1R-B*2704 C67S (5 × 10^5^ cells), or C1R cells (5 × 10^5^ cells) in MEM-α medium supplemented with sodium bicarbonate (1.5 g/L), 0.2 mM inositol, 0.02 mM folic acid, 0.01 mM 2-mercaptoethanol, 12.5% FBS, 12.5% horse serum, and 5% CO_2_ for one day at 37 ℃. To examine the effect of BH2—a monoclonal antibody that recognizes the α_2_-domain of B*27-HC and homo-oligomeric B*27-HC molecules [27]—on the perforin production of NK-92MI cells co-cultured with C1R-B*2704 cells, 20 μg of BH2 or mouse IgG were added to the medium during incubation. Cells were isolated via centrifugation (1200 rpm) for 3 min. The proteins in the pelleted cells were re-suspended with 1% SDS and ruptured via ultrasonication. An aliquot of the protein extract was analyzed through Western blotting and probed with anti-actin and anti-perforin antibodies.

### 2.4. Analysis of NK Cell-Mediated Cytotoxicity

NK-92MI cells (1 × 10^6^ cells) were co-cultured with C1R-B*2704 cells (5 × 10^5^ cells), C1R cells (5 × 10^5^ cells), or C1R-B*2704 C67S cells (5 × 10^5^ cells) in MEM-α medium supplemented with sodium bicarbonate (1.5 g/L), 0.2 mM inositol, 0.02 mM folic acid, 0.01 mM 2-mercaptoethanol, 12.5% FBS, 12.5% horse serum, and 5% CO_2_ for 4 h. Apoptotic cells of C1R-B*2704, C1R, or C1R-B*2704 C67S were analyzed through flow cytometry using double staining. Anti-CD19 antibodies conjugated with FITC and propidium iodide were used to stain B-lymphoblast cells (C1R-B*2704, C1R-B*2704 C67S, or C1R cells) and apoptotic cells, respectively (CD19 and propidium iodide kit, BD Biosciences).

### 2.5. Analysis of the Signaling Pathway of NK Cells Induced by Co-Culturing with C1R, C1R-B*2704, or C1R-B*2704 C67S Cells

NK-92MI cells were maintained in MEM-α medium supplemented with sodium bicarbonate (1.5 g/L), 0.2 mM inositol, 0.02 mM folic acid, 0.01 mM 2-mercaptoethanol, 12.5% FBS, 12.5% horse serum, and 5% CO_2_. NK-92MI cells (1 × 10^6^ cells) were co-cultured with C1R-B*2704 cells (5 × 10^5^ cells), C1R-B*2704 C67S (5 × 10^5^ cells), or C1R cells (5 × 10^5^ cells) in MEM-α medium at 37 ℃ for 10 min, in the absence or presence of the PI3K inhibitor (LY294002). Cells were withdrawn at the indicated time. NK-92MI cells were isolated by using the EasySep™ Human NK Cell Isolation Kit (STEMCELL), following the procedure recommended by the manufacturer, and then ruptured with 1% SDS. An aliquot of extracted proteins was separated via SDS-PAGE, analyzed through Western blotting, and probed by anti-phospho-AKT, anti-AKT, anti-perforin, and anti-actin.

### 2.6. Targeted Delivery of HLA-B27-Binding Peptide by THU Vehicle into the ER of C1R-B*2704 Cells and Analysis of the Perforin Production of NK-92MI Cells Co-Cultured with Peptide-Targeted C1R-B*2704 Cells

The sequence of the tat-derived peptide is GRKKRRQRRR. The tat-derived peptide can translocate the protein cargo across the cell membrane. KRGILTLKY and SRYWAIRTR derived from human actin and nucleoproteins of the influenza virus, respectively, are HLA-B*27-binding peptides [24,25]. THU is a fusion protein that contains one tat-derived peptide, histidine tag (His)_6_, and ubiquitin from N- to C-terminus. THU can be fused with KRGILTLKY and SRYWAIRTR to generate the fusion proteins THUNP and THUA, respectively. THU can deliver the HLA-B*27-binding peptides from the extracellular space into the cytoplasm of cells, where the antigenic peptides KRGILTLKY and SRYWAIRTR are cleaved from THUNP and THUA, respectively, by the cytosolic ubiquitin C-terminal hydrolases. The released antigenic peptides can be transported into the ER by the ER membrane proteins, the transporter associated with the antigen processing protein complex. The antigenic peptides in the ER can promote the folding of B*27-HC and suppress the production of B*27-HC oligomers [24,25]. THU, HUNP, HUA, THUA, and THUNP were prepared as described [24,25]. C1R-B*2704 cells (2 × 10^6^ cells) cultured in IMDM with 5% CO_2_, 10% FBS, and hygromycin B (200 μg/ml) (Invitrogen) were pre-treated with 20 μg of THU, HUNP, HUA, THUA, or THUNP for 24 h [24,25]. The cells were harvested via centrifugation. NK-92MI cells (1 × 10^6^ cells) were co-cultured with the pre-treated C1R-B*2704 cells (5 × 10^5^ cells) in a MEM-α medium supplemented with sodium bicarbonate (1.5 g/L), 0.2 mM inositol, 0.02 mM folic acid, 0.01 mM 2-mercaptoethanol, 12.5% FBS, 12.5% horse serum, and 5% CO_2_ for one day. Cells were harvested via centrifugation, resuspended in 1% SDS, and ruptured through ultrasonication. An aliquot of the protein extract was analyzed through Western blotting and probed with anti-actin and anti-perforin antibodies.

### 2.7. Quantitative Real-Time PCR

NK-92MI cells co-cultured with C1R-B*2704, C1R-B*2704 C67S, or C1R cells followed the methods as mentioned above. NK cells were isolated using the EasySep™ Human NK Cell Isolation Kit after co-culturing. The total RNAs of NK-92MI cells were isolated by using the QIAamp RNA Blood Mini kit (QIAGEN, GmbH, Hilden, Germany). IFN-γ, TNF-α, or IL-6 mRNA amplified with real-time PCR using a One Step SYBR Ex Taq qRT-PCR kit (TaKaRa, Shiga, Japan) followed the methods [28]. Relative expression levels of mRNA were calculated with the following equation: (40−threshold cycle [Ct], adjusted by the expression of 18S rRNA).

### 2.8. Statistical Analysis

Data are presented as means ± SD. Statistical significance between groups was assessed using the Mann–Whitney U test and *p*-values less than 0.05 were considered statistically significant.

## 3. Results

### 3.1. Replacement of Cys-67 with Ser Avoids the Oligomeric Formation of HLA-B*2704

B*2704-HC has the tendency to fold slowly in the ER. It can form homo-oligomers linked by a disulfide bond at Cys-67 during folding. We examined whether the replacement of Cys-67 with Ser can avoid the homo-oligomeric formation of B*2704-HC. C1R-B*2704 and C1R-B*2704 C67S stably overexpressed B*2704-HC and B*2704-HC C67S heavy chain, respectively. The proteins on the cell surface were labeled with biotin and then extracted in the presence of iodoacetamide to block the thiol group of cysteine. The biotinylated proteins were precipitated by avidin–agarose beads and analyzed with non-reducing SDS-PAGE. Figure 1A shows that B*2704-HC can form B27-HC oligomers. Based on the mobility of SDS-PAGE, B*27-HC oligomers contain dimer and tetramer (Figure 1A). Site-directed mutagenesis of Cys-67 by Ser blocks the oligomeric formation of B*27-HC. No B*27-HC C67S oligomers are formed, as evidenced through SDS-PAGE (Figure 1A). In addition, the level of the B27-HC monomeric form on the cell surface is similar to that of B*27-HC C67S (Figure 1A,B). The level of monomeric B*27-HC on SDS-PAGE is predominantly derived from the native form of HLA-B*2704/β_2_m/peptide that can be recognized by W6/32, indicating that the levels of assembled HLA-B*2704/β_2_m/peptide and HLA-B*2704 C67S/β2m/peptide displayed on the cell surface of C1R-B*2704 and C1R-B*2704 C67S cells, respectively, are similar. BH2 is a monoclonal antibody that can recognize the native heavy chain of HLA-B*2704/β_2_m/peptide and homo-oligomeric B*27-HC molecules [27]. The HLA-B*2704 on the cell surface was stained with BH2. Figure 1C,D show that the levels of B*2704-HC on the cell surface membrane are higher than those of B*2704-HC C67S, suggesting that the increased molecules are derived from the homo-oligomers of B*27-HC species.

### 3.2. The Perforin Production and Cytotoxicity of NK-92MI Cells were Promoted by Co-Culturing with C1R-B*2704 Cells

The previous study has indicated that the homo-oligomeric B*27-HC molecules are the ligand of KIR3DL2 displayed on the cell surface of NK cells [13,29]. Thus, we wanted to ask whether the homo-oligomeric B*27-HC molecules can affect the cytotoxicity of NK-92MI cells. The production of perforin will increase when the cytotoxicity of NK-92MI cells has been activated. Compared with co-culturing with C1R-B*2704 C67S cells, whether the perforin production of NK-92MI cells can be stimulated after co-culturing with C1R-B*2704 cells was examined. The perforin production was significantly increased when NK-92MI cells were co-cultured with C1R-B*2704 cells (Figure 2A,B), suggesting that the homo-oligomeric B*27-HC molecules can promote the perforin production of NK-92MI cells. In addition, the cytotoxicity of NK-92MI cells was also examined via flow cytometry. C1R cells, belonging to the B-lymphoblasts, and apoptotic cells can be stained with anti-CD19 and propidium iodide, respectively. Figure 2C,D indicate that the levels of apoptotic C1R-B*2704 cells are higher than those of C1R-B*2704 C67S when co-cultured with NK-92MI cells. The effect of different effector-to-target ratios on perforin expression and cytotoxicity of NK-92MI cells was examined. The results shown in Appendix A indicate that the different effector-to-target ratios do not alter the NK-mediated cytotoxicity.

Targeted delivery of the HLA-B*27-binding peptide into the ER can promote the folding of HLA-B*27 and suppress the formation of (B*27-HC)_2_: the major form of homo-oligomeric B*27-HC molecules [24,25]. Thus, we delivered the HLA-B*27-binding peptides KRGILTLKY and SRYWAIRTR into the ER of C1R-B*2704 to suppress the production of (B27-HC)_2_. The perforin production was reduced when NK-92MI cells were co-cultured with C1R-B*2704 cells that have been pre-treated with THUA or THUNP (Figure 2E,F). THU lacks the fused HLA-B*27-binding peptide at its C-terminus. Although THU can be translocated into the C1R-B*2704 cells by tat-derived peptides, it fails to promote HLA-B*2704 folding and suppress the (B*27-HC)_2_ formation. Apparently, THU cannot suppress the NK-mediated cytotoxicity toward C1R-B*2704 cells (Figure 2E,F). Both HUA and HUNP lack the tat-derived peptide sequence and fail to be translocated into the C1R-B*2704 cells. Treatment with HUA and TUNP is incapable of blocking NK-mediated target cell lysis (Figure 2E,F).

BH2 preferably binds to HLA-B* and HLA-C* rather than HLA-A*. The binding site is located within the α2 domain of native HLA-B27 HC and homo-oligomeric B*27-HC molecules [27]. Thus, we examined whether the perforin expression of NK cells was affected during the co-culturing of NK-92MI cells with CIR-B*2704 cells in the presence of BH2. Figure 2G,H indicate that BH2 can suppress the perforin expression of NK-92MI cells induced by co-culturing with CIR-B*2704 cells.

### 3.3. The PI3K/AKT Signaling of NK-92MI Cells were Activated by Co-Culturing with C1R-B*2704 Cells

Recruitment of PI3K to phospho-Tyr of ITIM in the cytoplasmic tail of KIR3DL2 has been reported, in turn activating the PI3K/AKT signaling of NK cells [21]. Thus, we analyzed whether the co-culture of NK-92MI cells with C1R-B*2704 or C1R-B*2704 C67S cells can activate the PI3K/AKT pathway. Figure 3A,B show that the PI3K/AKT signaling of NK-92MI cells can be activated by co-culturing with C1R-B*2704 or with C1R-B*2704 C67S cells. However, the level of the PI3K/AKT pathway activated by C1R-B*2704 cells is higher than that of C1R-B*2704 C67S cells. In the presence of the PI3K inhibitor (LY294002), activation of PI3K/AKT signaling by co-culturing with C1R-B*2704 cells in NK-92MI cells was significantly suppressed (Figure 3C,D) and the perforin expression was also down-regulated (Figure 3E,F).

### 3.4. The IL6 and TNF-α Expressions of NK-92MI Cells were Suppressed by Co-Culturing with C1R-B*2704 Cells

After activation, NK cells can enhance the secretion of several cytokines, such as IFN-γ and TNF-α [30]. We examined whether co-culturing with C1R-B*2704 or with C1R-B*2704 C67S cells can affect the expression of TNF-α, IL6, or IFN-γ of NK cells. NK-92MI cells were isolated after co-culturing with C1R, C1R-B*2704 C67S, or C1R-B*2704 cells. Compared with C1R and C1R-B*2704 C67S cells, co-culture with C1R-B*2704 cells can down-regulate the IL6 and TNF-α expressions of NK cells (Figure 4A–C), but it does not affect the expression of IFN-γ.

KIR3DL2 contains two ITIM motifs, membrane-proximal and membrane-distal ITIMs, in its cytoplasmic tail. Site-directed mutagenesis and yeast two-hybrid assay indicated that PI3K binds to the membrane-proximal ITIM motif [21]. The current data suggest that the homo-oligomeric B*27-HC molecules displayed by C1R-B*2704 cells bind to the KIR3DL2 receptor of NK-92MI cells, inducing PI3K to bind to the membrane-proximal phospho-ITIM motif of the receptor and activate the PI3K/AKT signaling to up-regulate the expression of perforin and promote NK-mediate cytotoxicity (Figure 5).

## 4. Discussion

NK cells play a critical role in the host defenses in the innate immune system. They kill the virus-infected cells or tumor cells but avoid destroying the normal self-cells [30,31,32,33]. This killing discrimination arises from NK cells that express the specified receptors displayed on their cell surface, recognizing MHC class I molecules expressed on the normal cells. MHC class I molecules bind to the specialized receptors of NK cells and inhibit NK-mediated target cell lysis. Target cells defectively expressing one or more MHC class I alleles are susceptible to NK-mediated cytotoxicity. The virus-infected cells alter the peptide pattern displayed by MHC class I alleles and tumor cells suppress the expressions of MHC class I molecules, accounting for both of which are sensitive to NK-mediated cell lysis [32,33]. Indeed, C1R cells that do not express HLA-A* and HLA-B* alleles are susceptible to cytotoxicity mediated by NK cells. Transfection with HLA-A*3, HLA-B*7, or HLA-B*58 molecules in C1R cells reverses the susceptibility of cell lysis mediated by NK cells [23]. In this study, C1R cells were transfected with HLA-B*2704 or HLA-B*2704 C67S alleles. Transfection of HLA-B*2704 into C1R cells resulted in the homo-oligomeric B*2704-HC molecules displayed on the cell surface (Figure 1A). The homo-oligomeric B*2704 C67S molecules were not found on the cell surface of C1R-B*2704 C67S cells (Figure 1A). Transfection of HLA-B*2704 or HLA-B*2704 C67S alleles into C1R did not inhibit cell lysis mediated by NK cells, but it promoted NK cell cytotoxicity (Figure 2C,D). This consequence is also reflected in the induced expression of perforin in NK cells co-cultured with C1R-B*2704 or C1R-B*2704 C67S cells (Figure 2A,B). The promotion of perforin expression in NK-92MI is dependent on the classical HLA-B*2704/β_2_m/peptide heterotrimer and the homo-oligomeric B*27-HC molecules. Suppression of the (B*27-HC)_2_ formation by targeting the HLA-B*27-binding peptide in the ER of C1R-B*2704 cells arises from the promotion of HLA-B*27 folding, resulting in reducing the perforin expression of NK-92MI cells (Figure 2E,F). All evidence revealed that the classical HLA-B27 complex or the homo-oligomeric B*27-HC molecules on C1R cells up-regulated the NK cell-mediated cytotoxicity.

MHC class I molecules on the target cells bind to the killer cell inhibitory receptors (KIR) of NK cells and in turn induce tyrosine phosphorylation to the ITIM domain in their cytoplasmic tails. The phospho-Tyr residue recruits different SH2 domain-containing proteins, in turn inducing the negative or positive signals for NK cells [21]. In the negative signal, the phospho-Tyr recruits SHP-1 or SHP-2 to catalyze the dephosphorylation reaction and block the NK cell-mediated cytotoxicity. The phospho-Tyr can also recruit PI3K to carry out the positive signal for the growth and survival of NK cells [21]. Recent studies have revealed that (B*27-HC)_2_ binds to KIR3DL2 of NK or CD4^+^ T cells, triggering the positive signal and promoting cell growth and survival [16,17,18]. This suggests that the phospho-Tyr of ITIM recruited the PI3K and activated the PI3K/AKT pathway for cell proliferation and activation of NK-mediated cytotoxicity upon the homo-oligomeric B*27-HC molecules binding to KIR3DL2 of NK cells. The PI3K/AKT signaling of NK-92MI cells was activated during co-culturing with C1R-B*2704 or C1R-B*2704 C67S cells (Figure 3A,B). The perforin expression was up-regulated by the PI3K/AKT signaling in NK-92MI cells, as evidenced by treatment with LY294002, a specific PI3K inhibitor, suppressing the perforin production (Figure 3C,D).

Compared with C1R cells, co-culture with C1R-B*2704 C67S or C1R-B*2704 cells did not affect the expression of IFN-γ of NK-92MI cells (Figure 4A). It is possible that the signaling for up-regulation of IFN-γ expression is not activated by the homo-oligomeric B*27-HC molecules binding to their receptors of NK cells. We also found that co-culture with C1R-B*2704 cells can suppress the IL6 and TNF-α expressions of NK cells (Figure 4B,C). Recent reports have indicated that NK-mediated cytotoxicity can be suppressed by IL-6 [34]. A decrease in IL-6 expression in turn promotes NK-mediated cytotoxicity. This consequence was also supported by the result that the perforin expression by NK cells was stimulated by co-culturing with C1R-B*2704 cells (Figure 2A). The enhanced NK cell-mediated cytotoxicity induced by the homo-oligomeric B*27-HC molecules displayed on the cell surface may contribute to the pathogenesis of AS.

## 5. Conclusions

HLA-B*2704 HC has the tendency to fold slowly and form the homo-oligomeric B*27-HC molecules that can be translocated onto the cell membrane, as evidenced by the homo-oligomeric misfolded B*27-HC displayed on the cell surface. The formation of B*27-HC oligomers was dependent on the linkage of the disulfide bond at Cys-67. The mutation of Cys-67 by Ser disrupted the formation of B*27-HC oligomers on the cell membrane. The co-culturing of C1R-B*2704 cells up-regulates NK cells to increase the perforin expression and augment their cytotoxic activities toward C1R-B*2704 target cells via activation of the PI3K/AKT signaling. Although the homo-oligomeric B*27-HC molecules bind to their receptors of NK cells and promote NK-mediated target cell lysis, these interactions cannot activate NK cells to up-regulate the expression of IFN-γ.

## Figures and Tables

**Figure 1 medicina-58-01411-f001:**
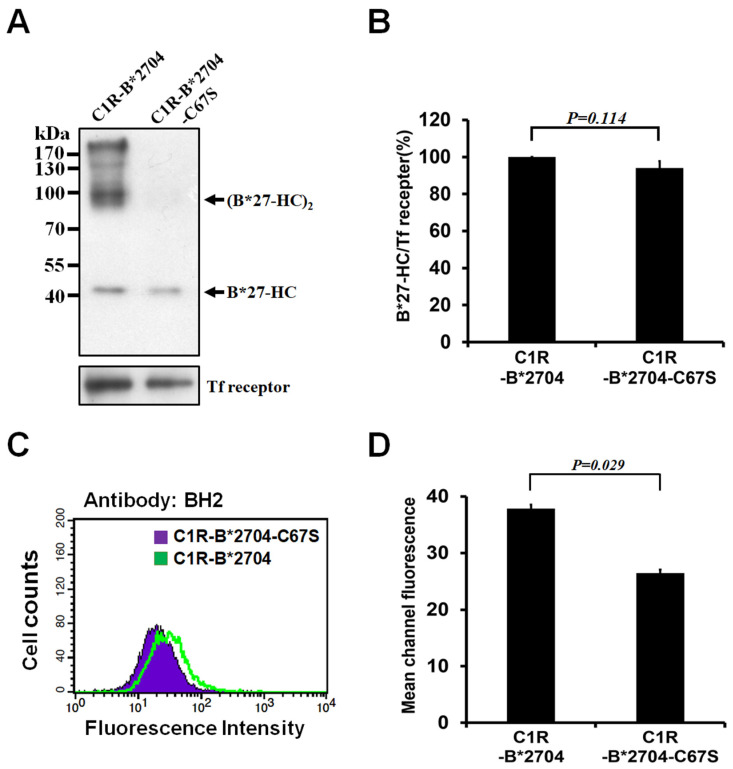
Analyses of membrane-bound B*27-HC species of C1R-B*2704 or C1R-B*2704 C67S cells through Western blotting and flow cytometry. The membrane-bound proteins of C1R-B*2704 or C1R-B2704 C67S cells were labeled with biotin and isolated by pull-down using avidin beads. (**A**) Analysis of membrane-bound B*27-HC species via Western blotting. An aliquot (50 μg) of each protein extract was separated using non-reducing SDS-PAGE, followed by Western blotting using the BH2 monoclonal antibody. (**B**) The ratio of monomeric B*27-HC/transferrin receptor obtained from Figure 1A is plotted. Values (mean ± SD, *n* = 4) are averaged from four independent experiments. (**C**) The membrane-bound HLA-B*2704 C67S or HLA-B*2704 was pre-associated with BH2, stained with FITC-conjugated secondary antibody, and analyzed via flow cytometry. (**D**) The results obtained in Figure 1C are plotted. Values (mean ± SD, *n* = 4) are averaged from four independent experiments.

**Figure 2 medicina-58-01411-f002:**
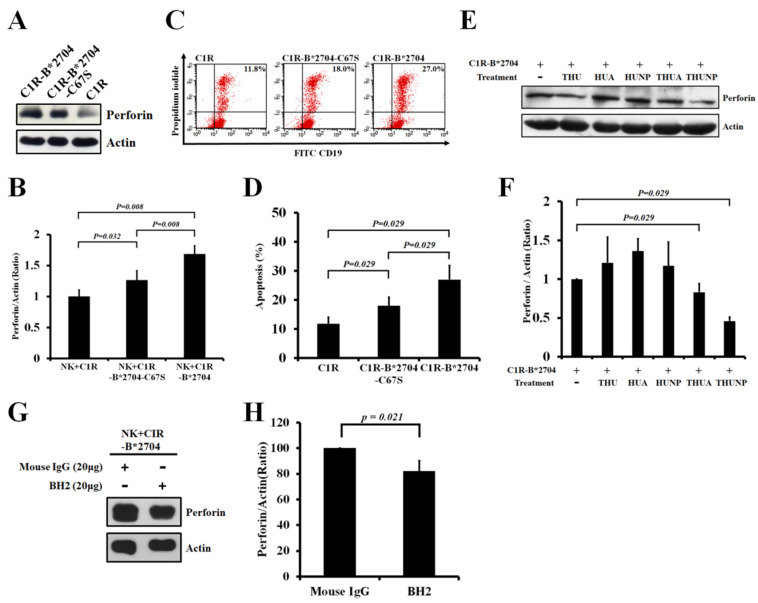
The effect of co-culturing with C1R, C1R-B*2704 C67S, or C1R-B*2704 cells on the NK cell-mediated cytotoxicity. (**A**) Western blotting analysis of the perforin expression in NK-92MI cells co-cultured with C1R, C1R-B*2704 C67S, or C1R-B*2704. An aliquot (50 μg) of extracted proteins was separated with SDS-PAGE and analyzed via Western blotting, probed for actin and perforin. (**B**) The ratio of perforin/actin averaged from four independent experiments in Figure 2A is plotted. (**C**) Analysis of NK-92MI-mediated cytotoxicity via flow cytometry. The apoptotic cells and C1R cells were stained with propidium iodide and anti-CD19 antibodies, respectively. (**D**) The percentages of apoptotic cells induced by NK-92MI-mediated cytotoxicity averaged from four independent experiments in Figure 2C are plotted. (**E**) The effect of co-culturing with C1R-B*2704 cells that have been pre-treated with THU, HUA, HUNP, THUA, or THUNP on the perforin expression of NK-92MI. (**F**) The ratio of perforin/actin averaged from four independent experiments in Figure 2E is plotted. The ratio of perforin/actin obtained from the control was set to one. (**G**). The effect of co-culturing with C1R-B*2704 in the presence of BH2 or mouse IgG on the perforin expression of NK-92MI. (**H**) The ratio of perforin/actin averaged from four independent experiments in Figure 2G is plotted. The ratio of perforin/actin obtained from the control was set to 100%.

**Figure 3 medicina-58-01411-f003:**
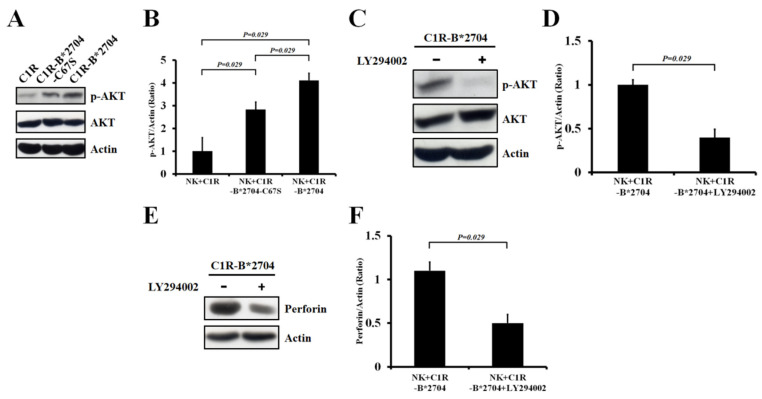
The effect of co-culturing with C1R, C1R-B*2704 C67S, or C1R-B*2704 cells on the PI3K/AKT signaling of NK-92MI cells. (**A**) Analysis of the PI3K/AKT signaling in NK-92MI cells induced by co-culturing with C1R, C1R-B*2704 C67S, or C1R-B*2704 cells by using Western blotting. An aliquot (50 μg) of crude proteins extracted from NK-92MI cells after co-culturing with C1R, C1R-B*2704 C67S, or C1R-B*2704 cells was separated with SDS-PAGE, analyzed via Western blotting, and probed for phospho-AKT, AKT, and actin. (**B**) The ratio of phospho-AKT/actin averaged from four independent experiments in Figure 3A is plotted. The ratio of phospho-AKT/actin obtained from NK-92MI cells co-cultured with C1R cells was set to one. (**C**) The induced PI3K/AKT pathway of NK-92MI cells co-cultured with C1R-B*2704 cells was blocked by the PI3K inhibitor. (**D**) The ratio of phospho-AKT/actin averaged from four independent experiments in Figure 3C is plotted. (**E**) The perforin expression of NK-92MI cells co-cultured with C1R-B*2704 was suppressed by treatment with the PI3K inhibitor. (**F**) The ratio of perforin/actin averaged from four independent experiments in Figure 3E is plotted.

**Figure 4 medicina-58-01411-f004:**
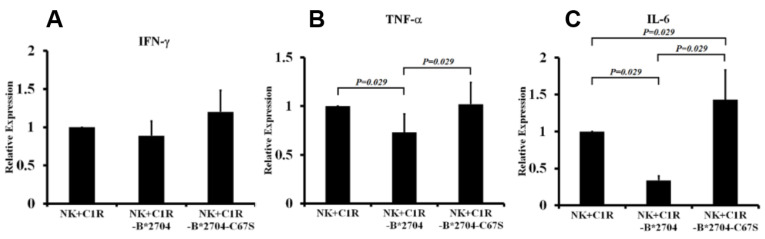
The effect of co-culturing with C1R, C1R-B*2704 C67S, or C1R-B*2704 cells on the cytokine expression of NK-92MI cells. After co-culturing with C1R, C1R-B*2704, or C1R-B*2704 C67S, NK-92MI cells were isolated by using the EasySep™ Human NK Cell Isolation Kit. The total RNAs of treated NK-92MI cells were isolated by using the QIAamp RNA Blood Mini kit (QIAGEN, GmbH, Germany). The results were averaged from four independent experiments. The levels of IFN-γ (**A**), TNF-α (**B**), or IL-6 (**C**) mRNA of NK cells were analyzed through quantitative RT-PCR. Real-time RT-PCR values for each cytokine were normalized to those of 18S rRNA. Relative expression levels of mRNA were calculated by the following equation: (40−threshold cycle [Ct], adjusted by the expression of 18S rRNA).

**Figure 5 medicina-58-01411-f005:**
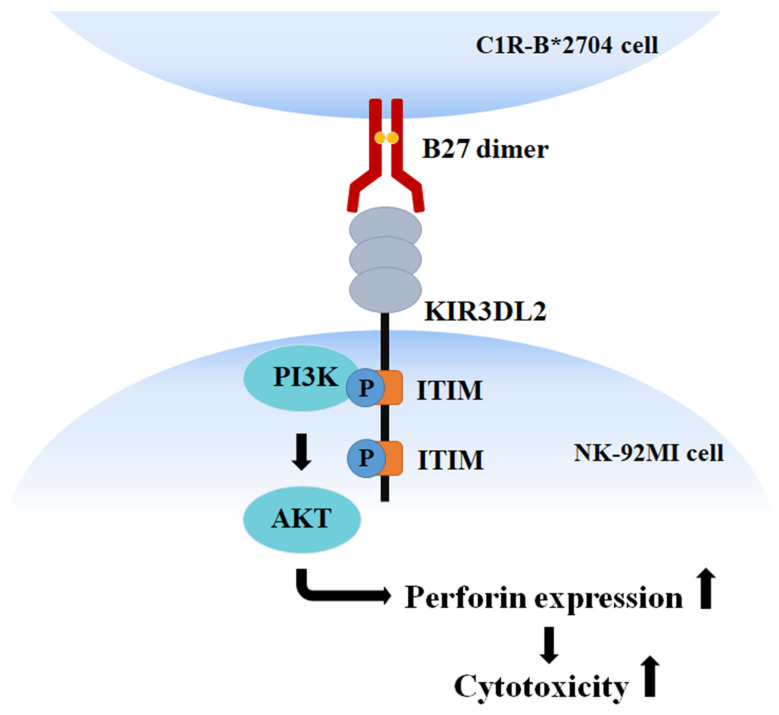
Schematic diagram illustrates the mechanism of the homo-oligomeric B*27-HC molecule to up-regulate the NK-92MI-mediated cytotoxicity.

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
