# Peer review of "HLA-B*27 Heavy Chain Homo-Oligomers Promote the Cytotoxicity of NK Cells via Activation of PI3K/AKT Signaling"

_medicina, 2022, doi:10.3390/medicina58101411_

Round 1

Reviewer 1 Report

Dear Authors,

It is a big pleasure and great honor to make a peer-review for your prestigious journal. In my opinion, it is a very good article, well written, novelty with a high impact which can be published after a minor revision. Please also add more conclusions to your article.

Kind regards

Author Response

We have followed the reviewers’ comments to amend our manuscript. The amended regions are marked by yellow in the amended manuscript. The review’s questions were answered as follows:

Reviewer 1:

It is a big pleasure and great honor to make a peer-review for your prestigious journal. In my opinion, it is a very good article, well written, novelty with a high impact which can be published after a minor revision. Please also add more conclusions to your article.

Ans: We have followed the reviewer’s suggestion to amend the conclusion.

Reviewer 2 Report

The authors present their results on the role of HLA-B*27 on NK cells' activity. However, I find that there are some major issues that need to be addressed and modified:

- The use of the terms "HLA-B*2704" and "HLA-B*2704 C67S" should be described better in the methods section.

- The cell culture methodology of C1R and NK92-MI cells needs to be added.

- The authors need to justify the use of different effector-to-target ratios in the cytotoxicity and signaling pathway experiments.

- The authors claim that signals through KIR3DL2 would be inhibitory but the word "activation" is used throughout the manuscript. Instead of activation, the authors should explain that it is the removal of the inhibitory signal leading to the activation of NK cells.

- In figure 1 legend, the reference to figure 1E is incorrect and needs to be replaced.

- The authors use primary isolated NK cells for the PCR experiments for IL-6 and TNF alpha. There is no rationale for the use of primary cells in only one experiment as it is known that the behavior of NK92MI and primary cells are different.

- The discussion needs major re-writing as it is a repetition of the results.

- Also, a summary conclusive figure might be helpful in summarizing the data reported by the authors. 

Author Response

We have followed the reviewers’ comments to amend our manuscript. The amended regions are marked by yellow in the amended manuscript. The review’s questions were answered as follows:

Reviewer 2:

  1. The use of the terms "HLA-B*2704" and "HLA-B*2704 C67S" should be described better in the methods section.

Ans: We have followed the reviewer’s suggestion to amend our manuscript.

  1. The cell culture methodology of C1R and NK92-MI cells needs to be added.

Ans: We have followed the reviewer’s suggestion to amend our manuscript.

  1. The authors need to justify the use of different effector-to-target ratios in the cytotoxicity and signaling pathway experiments.

Ans: We have done more experiments to examine the effect of different effector-to-target ratios in the cytotoxicity. The results are summarized in supplementary Figure 1. The increased ration of NK-92MI cells to C1R, C1R-B*2704 or C1R-B*2704 C67S cells did not alter the NK-mediated cytotoxicity.

  1. The authors claim that signals through KIR3DL2 would be inhibitory but the word "activation" is used throughout the manuscript. Instead of activation, the authors should explain that it is the removal of the inhibitory signal leading to the activation of NK cells.

Ans: We have modified the contents of Discussion to clearly explain the mechanism for this change from inhibition to activation.

  1. The authors use primary isolated NK cells for the PCR experiments for IL-6 and TNF alpha. There is no rationale for the use of primary cells in only one experiment as it is known that the behavior of NK92MI and primary cells are different.

Ans: We have corrected our errors in the revised manuscript. We used NK-92MI cell line for all studies. The rationale for these studies is added in the content of revised manuscript.

  1. The discussion needs major re-writing as it is a repetition of the results.

Ans: We have followed the reviewer’s suggestion to amend our manuscript.

  1. A summary conclusive figure might be helpful in summarizing the data reported by the authors.

Ans: We have summarized our findings in Fig. 5.

Round 2

Reviewer 2 Report

The authors addressed most of the comments. However, some chnages are still needed for the manuscript before publication.

1- Unify the use of symbols for interferon (IFN or INF).

2- Suppl Figure 1 has the figure that uses the same effector to target ratio for all 3 cell types. So it seems that there is a missing figure there?

3- Since the authors decided to remove work by primary NK cells and used NK92MI instead, what is the rationale for linking this to AS in particular, not other associated autoimmune diseases?

 4- The data presented in figure 4 is the exact same compared to the first submitted figure using primary NK cells. How is that possible?

Author Response

We have followed the reviewers’ comments to amend our manuscript. The amended regions are marked by yellow in the amended manuscript. The review’s questions were answered as follows:

1- Unify the use of symbols for interferon (IFN or INF).

Ans: We have unified the abbreviation of interferon-g for IFN-g.

2- Suppl Figure 1 has the figure that uses the same effector to target ratio for all 3 cell types. So it seems that there is a missing figure there?

Ans: NK-92MI cells grow very slowly. We couldn't finish all experiments within the deadline. So, the experiments of signaling pathway affected by different effector-to-target ratios were not finished within the deadline. But, these experiments have been done now. The results are shown in Figure 1.

Figure 1. The effect of co-culture with C1R, C1R-B*2704 C67S or C1R-B*2704 cells on the PI3K/AKT signaling of NK-92MI cells. NK-92MI cells (1 x 106 cells) were co-cultured with C1R-B*2704 cells (2 x 105 cells), C1R-B*2704 C67S (2 x 105 cells) or C1R cells (2 x 105 cells) in MEM-a medium supplemented with sodium bicarbonate ( 1.5 g/L), 0.2 mM inositol, 0.02 mM folic acid, 0.01 mM 2-mercaptoethanol, 12.5% FBS , 12.5% horse serum and 5% CO2 at 37℃ for 10 min. NK-92MI cells were isolated by using the EasySep™ Human NK Cell Isolation Kit (STEMCELL), following the procedure recommended by the manufacturer, and then ruptured by 1% SDS. (A). Analysis of the PI3K/AKT signaling in NK-92MI cells induced from co-culture with C1R, C1R-B*2704 C67S or C1R-B*2704 cells by using western blotting. An aliquot (50 mg) of crude proteins extracted from NK-92MI cells after co-culture with C1R, C1R-B*2704 C67S or C1R-B*2704 cells, was separated by SDS-PAGE and analyzed by western blotting, probed for phospho-AKT, AKT and actin. (B) The ratio of phospho-AKT/actin averaged from four independent experiments in Figure 1A is plotted. The ratio of phospho-AKT/actin obtained from NK-92MI cells co-cultured with C1R cells was set to 1.

3- Since the authors decided to remove work by primary NK cells and used NK92MI instead, what is the rationale for linking this to AS in particular, not other associated autoimmune diseases?

Ans: NK cells are only a small fraction in human PBMCs. Thus, human NK cell line is always used to replace the primary NK cells in AS studies. The homo-oligomeric HLA-B*2704 molecules binding to the KIR3DL2 receptor of NK cells are only found in AS. This interaction is not associated with other autoimmune diseases.

 4- The data presented in figure 4 is the exact same compared to the first submitted figure using primary NK cells. How is that possible?

Ans: We never used the primary NK cells in these studies. We corrected our errors in the revised manuscript. Thus, the figure 4 of the revised version is the same as the part of figure 4 in the original manuscri
